# Identifying the Configuration Differences of Primary Schools with Different Administrative Affiliations in China

**Wenwen Sun [1,2], Xin Hu [2], Zhuoran Li [1] and Chunlu Liu [2,*]** 

[1]  School of Architecture and Urban Planning, Shandong Jianzhu University, Jinan 250101, China; sunwenw@deakin.edu.au (W.S.); 12988@sdjzu.edu.cn (Z.L.)

[2]  School of Architecture and Built Environment, Deakin University, Geelong 3220, Australia; xin.hu@deakin.edu.au

[\*]  Correspondence: Chunlu.Liu@deakin.edu.au

**Abstract:** Equalization of education facilities, which means the balanced distribution of human and material resources under limited resource conditions, is one of the goals of sustainable development. In the process of rapid urbanization in China, there are apparent discrepancies between urban and rural areas because of different land and household registration systems. Primary schools with three types of different administrative affiliations also have significant distinctions. This study is aimed at assessing and comparing the configuration of primary schools with three different administrative affiliations, including cities, towns, and villages. After building an indicator system, the entropy weight method is applied to calculate the overall and category configuration scores of each school. Based on a spatial database, the ArcGIS thematic maps display the geography characteristic of each school in different geolocations. Moreover, the Kruskal–Wallis test identifies if the configuration of primary schools with different administrative affiliations is equal. The results indicate that the allocation of primary schools with three different administrative affiliations showed a ternary development. Moreover, although primary schools with city affiliation had significant advantages in education quality and convenience, their supply–demand conditions were not optimistic. In addition, the quality of the primary schools subordinate to towns was better, but convenience was generally lower. Finally, the quality of primary schools subordinate to villages and some towns was still poor. The results provide facts and a basis for policymakers to achieve an equity configuration in the sustainable development context.

**Keywords:** configuration; primary schools; sustainable development; entropy weight method

## 1. Introduction

China, whose urbanization rate was 59.58% in 2019, is still in a rapid urbanization development process in most cities [1]. According to the statistics, there is an apparent differentiation in the administrative boundary, economics, and society among cities, towns, and villages, and cities' economic development is better than towns' and towns' is better than villages'. The configuration of education facilities is influenced by this economic development, more or less [2]. For a long time, the countryside has been a shortcoming of the education development in China. The population is increasingly migrating from villages to towns and cities. The government expects towns could play an important role in stranding the urbanization population to reduce the population and environmental pressures in large cities. Meanwhile, an equal education environment for all citizens is also an important goal of the Chinese government. Moreover, the government has proposed a large number of laws [3], policies [4,5], and specifications [6] to promote and achieve that aim over the past decades. However,

whether the configuration of education facilities is fair or not always depends on many objective factors. If there is no statistical analysis, such a fair configuration will be difficult to confirm even only among three different administrative affiliations, which are the three levels of administrative divisions—cities, towns, and villages in China.

Equal access to educational public goods and quality services for all is one of the sustainable development aims [7]. Sustainable development should ensure inclusive and equitable quality education for all, especially in the process of dynamic urbanization [8]. By 2030, the Chinese government will attempt to ensure that all boys and girls gain free, fair, and quality primary and secondary education and realize equalization of basic public educational services in urban and rural areas. [9]. It is significant to identify the configuration difference of primary schools with three different administrative affiliations to present strategies and advanced approaches in sustainable urban development.

The studies associated with education facilities last a long time. Researchers in different countries at different stages of urbanization development have different focuses. Sociological scholars focused on the quantitative study of school segregation with much emphasis placed on ethnic and racial dimensions, mainly in America and Europe, including the impact on home-to-school mobility, the differences in charters, and public and private schools [10–14]. School selection is a common problem, existing in many counties, that involves parental choices of schools, school choice mechanisms, impact, and policy and development trends [15–19]. There are also a series of studies associated with commuting to school, including accessibility and the mode of transportation [20,21].

The balanced development of education facilities is the common goal pursued by various countries. Previous research on the inequity of education services has mainly focused on different income classes, education reform, accessibility, and facility maintenance and investment [22–25]. There are also some studies focusing on school performance issues [26]. Although a few scholars focus on the study of rural education facilities, there are few quantitative studies on urban–rural differences [27,28]. The purpose of this research is to determine whether there are differences in the configuration of primary schools with three different administrative affiliations, and which aspect these differences are mainly reflected in.

## 2. Basic Education Facilities and Sustainable Development

Public economics authorities Atkinson and Stiglitz viewed basic education as publicly available private products [29]. The famous American policy scientist Inge Kaur also believes that the free provision of basic education and the realization of publicity are not due to the non-exclusiveness and non-competitiveness, but because of the results of government institutional arrangements [30]. Basic education facilities, which are essentially private products, could turn into the public products by special measures. Many countries have formed education quasi-markets through the introduction of private education and the reform of school selection mechanisms. However, except for pre-school education institutions, basic education facilities have obvious public goods attributes in China because more than 90% of basic education belongs to public schools supplied by public finances [31]. Thus, basic education facilities are subject to public policies and belong to public service facilities. A small number of private schools are not in the research scope of this article.

Primary schools are an important part of basic education facilities. Primary schools, limited by children's school age, have strict requirements for the distance to school. Moreover, the number of primary schools, constrained by the running scale, is significantly higher than that of secondary and high schools, so it is more typical. Educational economics believes that the configuration of educational resources refers to how to allocate limited educational resources among all education level, regions, and schools so that the educational resources invested can be fully and effectively used. In China, the configuration of basic education facilities is mainly implemented through two departments. The urban planning department is mainly responsible for determining the schools' land and building area according to the size of the population, and then conducting the school layout

according to different levels of the community-life circle; the education department is responsible for education funding and teacher allocation. In this paper, the configuration of primary schools refers to the rational and reasonable allocation and layout of teachers, land, and building resources. The equity configuration of primary schools refers to the balanced distribution of human resources and school land and building resources under limited resource conditions, which is also the basic requirement for achieving sustainable development.

The balanced development of public goods is the goal of sustainable development. Providing basic conditions for sustainable living, protecting and satisfying the basic needs of the most vulnerable people in society, and providing equal public services for all people are important part of sustainable development [32]. Narrowing the difference in the supply of urban and rural public goods and providing a balanced and equitable institutional and financial supply are the inevitable ways to achieve sustainable urban and rural development. A widening gap in public services will cause a series of social contradictions and hinder the process of urbanization. Accessing high-quality education service is the foundation for improving people's lives and achieving sustainable development. Prioritizing the development of education has been a major policy that China has long adhered to. In recent years, China's central and local governments have been working hard to promote the balanced development of the urban and rural basic education service system, continuing to increase the supply of basic education products in rural areas and investment in rural education [33]. The balanced and sustainable development of education should be implemented in consideration of the actual capabilities, development levels, resources, and infrastructure of different countries. Different countries should formulate education indicators and planning strategies suitable for national development in accordance with sustainable development strategies. China is in the acceleration stage of urbanization. The population mobility in urban and rural areas has made it more difficult for the balanced development of public services.

## 3. Methodology

This paper developed a methodology to confirm there is a difference among the configuration of primary schools with three different administrative affiliations in China. The confirmation of the difference identified the existence of the problems and built an aim and direction to improve sustainable urban. The procedures for measuring the difference among primary schools are given as follows. First, the performance of primary schools is evaluated through the entropy weight method; thereby, the ArcGIS 10.7.1 (Environmental System Research Institute (ESRI), Esri Australia Pty. Ltd., Brisbane, Australia) thematic maps display the spatial characters result from the entropy weight method [34,35]. Finally, the Kruskal–Wallis test of multiple independent samples is used to confirm the difference among three different type primary schools through the scores. This method can also be used for other case studies, for example, the comparison of school configurations in Eastern and Western China.

### 3.1. The Value Calculation of Primary Schools through the Entropy Weight Method

The entropy method [36] is one of the classic algorithms for calculating index weights, and it refers to a mathematical method used to judge the degree of discreteness of an index. The greater the entropy value is, the greater the disorder degree of the system. The entropy weight method, which is one of the objective fixed weight methods, determines the index's weight based on the amount of information. The entropy method is used to calculate the weight of the indicators in this paper.

As all the indicators have different magnitudes and dimensions, normalization is needed for converting all indicators into similar measurement scales. First, extreme values should be removed. A Min-Max rescaling method is used to normalize the positive and negative indicators with the following Equation (1). The positive indicators mean the larger the value, the better the performance, while for the negative indicators, a larger value represents better performance.

$$
T_{ij} = \begin{cases} \dfrac{X_{ij} - \underset{i=1}{\overset{n}{MIN}}(X_{ij})}{\underset{i=1}{\overset{n}{MAX}}(X_{ij}) - \underset{i=1}{\overset{n}{MIN}}(X_{ij})} & X_{ij} \text{ is positive indicator} \\[4mm] \dfrac{\underset{i=1}{\overset{n}{MAX}}(X_{ij}) - X_{ij}}{\underset{i=1}{\overset{n}{MAX}}(X_{ij}) - \underset{i=1}{\overset{n}{MIN}}(X_{ij})} & X_{ij} \text{ is negative indicator} \end{cases} \tag{1}
$$

where $X_{ij}$ is the original value of the indicator $i$ for sample primary school $j$, and $T_{ij}$ is the normalized value of the indicator $X_{ij}$. If $X_{ij}$ is an optimal indicator, it should be normalized by Equation (2):

$$
T_{ij} = \begin{cases} 1 - \dfrac{|X_{ij} - X_t|}{\left| \underset{i=1}{\overset{n}{MIN}}(X_{ij}) - X_t \right|} & if\ X_{ij} < X_t \\[6mm] 1 - \dfrac{|X_{ij} - X_t|}{\left| \underset{i=1}{\overset{n}{MAX}}(X_{ij}) - X_t \right|} & if\ X_{ij} \geq X_t \end{cases} \tag{2}
$$

where $X_{ij}$ represents the original data, $T_{ij}$ represents the dimensionless data after normalization, and $X_t$ is the optimal threshold' values. The determination of the optimal threshold' values needs to be combined with the local standards and actual conditions in different regions. Assume there are $m$ indicators for configuration of $n$ sample of primary schools. $T_{ij}$ is the normalized value of indicators defined in Equations (1) and (2). First, the normalized data is used to calculate the proportion $P_{ij}$ of the school $j$ in the indicator $i$ with Equation (3):

$$
P_{ij} = \frac{T_{ij}}{\sum_{i=1}^{n} T_{ij}} \quad (i = 1, 2, \ldots m; j = 1, 2, \ldots n) \tag{3}
$$

According to the definition of entropy, the entropy of the $i$th indicator in the $j$th primary school can be calculated with Equation (4):

$$
H_i = -\frac{\sum_{i=1}^{n} P_{ij} ln(P_{ij})}{lnn} (i = 1, 2, \ldots m; j = 1, 2, \ldots n) \tag{4}
$$

Entropy weight of the $i$th indicator is determined by Equation (5):

$$
W_i = \frac{1 - H_i}{m - \sum_{i=1}^{m} H_i} \sum_{i=1}^{m} W_i = 1, (i = 1, 2, \ldots m) \tag{5}
$$

where $H_i$ represents the entropy of the indicator $i$, and $m$ is the total number of indicators. The overall score and category score of each school $j$ are calculated with Equation (6):

$$
S_j = \sum_{i=1}^{m} W_i \times P_{ij} \ (i = 1, 2, \ldots n; j = 1, 2, \ldots n) \tag{6}
$$

*3.2. Measurement of Differences in Primary School Configurations*

The Kruskal–Wallis test of multiple independent samples, which is a non-parametric method for testing whether samples originate from the same distribution, is used for comparing two or more independent samples of equal or different sample sizes. It is applied for measuring the difference of the configuration of primary schools with three different administrative affiliations in China in this paper. The Kruskal–Wallis test is based on the propose assumption: There is no difference among the

configuration of primary schools with three different administrative affiliations. The test statistic is given by Equation (7):

$$H = \frac{12}{N(N+1)} \sum_{i=1}^{k} n_i \left( \overline{R}_i - \overline{R} \right)^2 \tag{7}$$

where $n_i$ is the sample size of group $i$, $N$ is the total sample size of all groups, $\overline{R}_i$ is the mean rank of group $i$, and $\overline{R}$ is the mean rank, where $\overline{R} = \frac{N+1}{2}$. As for the test results, the test statistic, which is the value of $H$, proves the adequacy of the null hypothesis. The probability, which is the value of the statistical significance of the test, decide whether the null hypothesis is rejected. If it is less than 0.05, the original hypothesis is rejected, and there is a significant difference. The probability is greater than 0.05, the original hypothesis is accepted, and there is no significant difference.

## 4. Case Study Description

### 4.1. Study Area and Data Sources

Zibo is a group city, second only to Jinan and Qingdao in terms of area and urbanization rate in Shandong Province, under the jurisdiction are five districts and three counties. The districts include Zhangdian, Zichuan, Boshan, Zhoucun, and Linzi District. The counties contain Huantai, Gaoqing, and Yiyuan. The research scope of this article is the administrative scope of Zibo City. Primary schools in Zibo City are the research object of this paper. The statistics in this article, which contain 384 primary schools, take into account the nine-year compulsory education schools and rural learning centers together excluding private schools. The scope of the paper in the article is based on the administrative division map and a Google map of Zibo City. The scope of the school district is based on the related information and surveys published on the Zibo Education Information Website in 2018. The other relevant data used in the article are all from field surveys and the Internet.

According to administrative divisions and affiliation, primary schools in Zibo can be divided into urban primary schools, town primary schools, and village primary schools. Generally, the department, which the primary school is affiliated with, is consistent with the main bodies of financial input. There are 161 village primary schools, 124 town primary schools, and 99 urban primary schools. The administrative affiliation classification of the primary schools is corresponding to the spatial location of the schools, and a database is established to obtain the spatial distribution map of primary schools through ArcGIS 10.7 [37–39], as shown in Figure 1. Primary schools subordinated to cities, towns, and villages are displayed in blue, yellow, and brown, respectively. It is obvious from Figure 1 that the urban schools are densely distributed and their school districts are much smaller than these two other categories, while some town and village schools always serve a large area. Although some village schools also serve a small area, the layout of these small village schools is relatively scattered. This is because the city, town, and village communities served by them are quite different from each other in volume rate, building density, and population density.

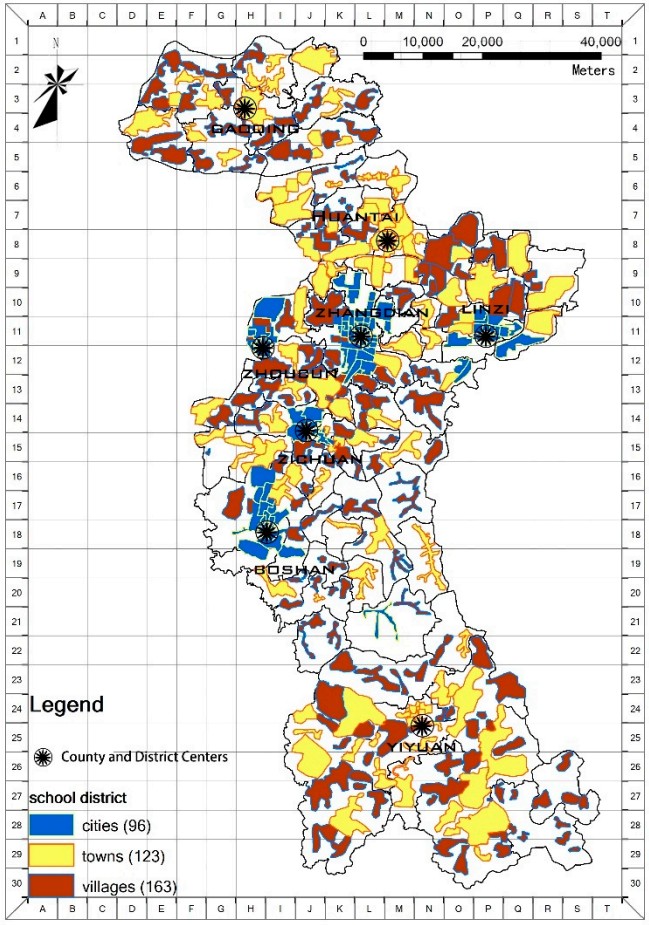

**Figure 1.** The administrative affiliation classification of primary schools' distribution in Zibo City.

*4.2. Selection of the Indicators for the Configuration of Primary Schools*

The selection of the indicators for the primary schools' configuration mainly depends on two factors. One is the literature review, and the other is the national norms and standards restricting the configuration of primary schools in China. Due to the relatively complete development of education facilities in these countries, there is generally no need for large-scale demolition and construction, so few articles systematically summarize the index system of the education facility configuration from the perspective of planning and construction. Moreover, previous evaluation indicators for schools primarily concentrated on the function and performance, but physical and environmental indicators are less involved [26,40]. However, the indicators appearing in some literature also show that they have typical characteristic reflecting school configuration. The main factor for the primary school configuration is the balance between the supply and demand of resources. China national standards' main control elements of school supply are the land and building area of the school. At the same time, referring to some literature, the human factors presented by the teacher–student ratio should also be included. Moreover, the quality of education facilities largely determines the choice of residents. China's constraints on the quality of primary schools mainly rely on the standardized educational evaluation. The evaluation results characterized by teaching management quality indicators have significant reference values. In addition, a higher-education teacher–student ratio is involved, referring to relevant literature. In order to ensure residents' use convenience, the four indicators, including transportation distance, transportation time, 500-m radius coverage, and the number of residential sites served are included, after referring to the literature and specifications.

Specifically shown in Table 1, supply and demand indicators are about the student-to-resource ratio. Quality indicators refer to teacher quality and comprehensive evaluation of school quality. The

latter one is determined by the results of the standardized school education evaluation. Indicators for convenience are mainly related to transportation profiles and service scope of primary schools. Primary indicators of the three major categories for assessments are summarized in Table 1.

**Table 1.** Primary indicators and weight value for primary schools in major categories.

| Category | Category Weights | Indicators | Effect | Justification | Indicator Weights |
|---|---|---|---|---|---|
| Supply and Demand | 0.3133 | Teacher-student ratios | Optimal | [6,40,41] | 0.0865 |
| | | Land area per student | Optimal | [42] | 0.0895 |
| | | Building area per student | Optimal | [40–44] | 0.1373 |
| Quality | 0.3014 | Highly educated teacher-student ratios | Optimal | [45] | 0.0794 |
| | | Teaching management quality index | Positive | [40] | 0.2220 |
| Convenience | 0.3853 | Actual traffic distance | Negative | [46] | 0.0712 |
| | | Actual traffic time | Negative | [47] | 0.0492 |
| | | 500 m radius coverage ratio | Positive | [41,42,48] | 0.2243 |
| | | Number of residential settlements served | Negative | [49] | 0.0406 |

For each school, the teacher–student ratio and highly-educated teacher–student ratio refers to the ratio of teachers or teachers with Bachelor and Master's Degree to students, respectively. Land area per student presents the ratio of total land area to students. The 500 m radius coverage ratio mean the ratio of the area of circle with a radius of 500 m centered on the school to the total area of the school district. Teaching management quality index, an arithmetic progression, is established according to the provincial and municipal levels of the standardized school whose tolerance is equal to 5 and the time batch of the standardized school whose tolerance is equal to 1. Actual traffic distance and time are estimated as the average traffic distance or time of every family in a school district, respectively. The number of residential settlements refers to the number of residential settlements served by a school.

The value in Table 2 refers to China's national standards and specifications about schools and take into account the actual situation of primary schools in Zibo City. The standards and specifications are shown in Table 1. This also refers to the statistical data of Educational Statistics Yearbook of China 2018 [31].

**Table 2.** A list of the optimal threshold' values.

| Optimal Indicator | Student-Teacher Ratios | Land Area Per Student | Building Area Per Student | Highly Educated Teacher-Student Ratios |
|---|---|---|---|---|
| optimal values | 14:1 | 40 | 16 | 14:1 |

According to the index system of the above Section 3.1 and the calculation method of Section 3.2, the weight values of different classifications and specific indicators are shown in Table 1. Three categories, including supply and demand, quality, and convenience, have similar weight values, ranging between 0.3 and 0.4. Among them, the convenience category whose weight value is the highest contains four specific indicators. By contrast, the quality category whose weight value is the lowest involves only two specific indicators. For specific indicators, "500 m radius coverage ratio" has the most significant impact on the final evaluation results, closely followed by "teaching management quality index", while the two lowest impact indicators are "number of residential settlements served", and "actual traffic time".

## 5. Results

*5.1. The Results of the Entropy Weight Method*

After obtaining the weight value for each indicator, the overall and category scores of 384 primary schools are calculated by Equation (6). To observe the profiles of all primary schools' evaluations, the scatter plot of primary schools' comprehensive scores is illustrated in Figure 2. The full sample data of primary schools in Zibo City is divided into three categories to display the difference among them. The value of the *x*-axis is meaningless. The evaluation results show that city primary schools score higher than towns and villages.

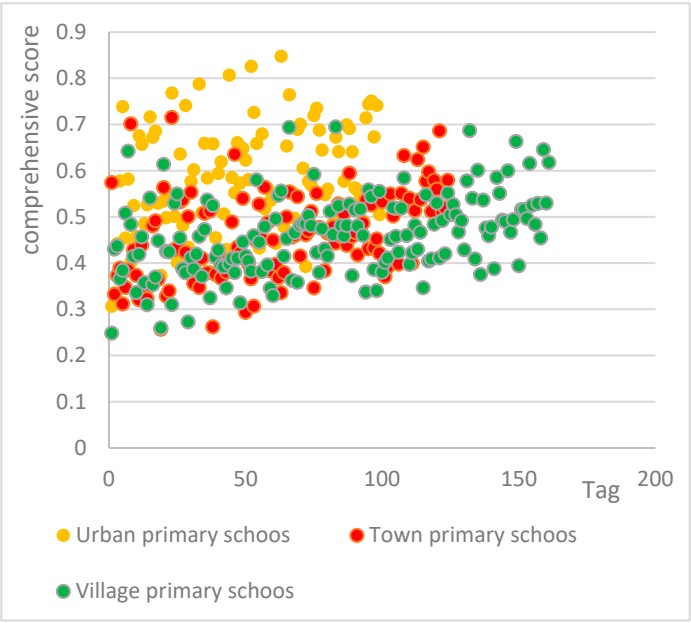

**Figure 2.** The scatter plot of primary schools' comprehensive score in Zibo City.

However, the distribution profile of primary schools' evaluation results could still not come to light only by observing the scatter plot. Specifically, for the primary schools with three different administrative affiliations, it is unknown where the primary schools' scores are high and where the primary schools' scores are low. Therefore, we correspond the overall and categories scores with the location of the schools, and establish a data connection in ArcGIS 10.7 to obtain the spatial distribution map of different primary schools' configuration, as shown in Figures 3 and 4. The color changing from blue to red indicates the scores rising from low to high. School district outlines indicate primary schools with different affiliation classification.

Figure 3 demonstrates the spatial distribution profile of the three classification scores. Surprisingly, from Figure 3a, it is seen that the supply and demand scores in urban areas have a disadvantage, while the supply and demand profiles of town and village primary schools are relatively good. There are 18 primary schools scored below 0.1, which show that they have poor supply and demand conditions. Of them, two are village schools, five are town schools and 11 are urban schools. A total of 225 primary schools' supply and demand scores are between 0.1 and 0.2. Of them, 60 are urban primary schools, 74 are town primary schools, and 91 are village primary schools. Additionally, 140 primary schools score between 0.2 and 0.3. Of them, there are 68 village primary schools, 44 town primary schools, and 28 urban primary schools. Only one primary school scores beyond 0.3, which is a town primary school having the best supply and demand conditions.

The profile of primary schools' quality scores is demonstrated in Figure 3b. Interestingly, in addition to the well-known urban primary schools, some township and village primary schools also score high. There are 62 primary schools that scored between 0.2 and 0.3, which shows they have a

relative advantage in terms of school quality. Of them, the number of urban primary schools is 34, township primary schools' is 18, and village primary schools' is 10. The scores of 180 primary schools are between 0.1 and 0.2. Of them, there are 71 village schools, 50 township schools, and 59 urban schools. Importantly, village primary schools still take up over half the number of primary schools getting the lowest scores. A total of 142 primary schools score below 0.1, which mean they have a relative disadvantage in terms of school quality. Of them, there are six urban primary schools, 56 town primary schools, and 80 village primary schools.

Figure 3c illustrates the profile of primary schools' convenience scores. Undoubtedly, urban primary schools with the smallest school district have the most convenient conditions of school-to-home commute. A total of 39 schools score over 0.3, which demonstrate these schools are good with respect to convenience conditions. Of them, there are 32 urban primary schools, two township primary schools, and five village primary schools. A total of 100 schools score between 0.2 and 0.3. Of them, the number of urban schools is 37, the number of town schools is 26, and the number of village school is 37. A total of 212 schools score between 0.1 and 0.2. Of them, 28, 82, and 102 schools belong to cities, towns, and villages, respectively. Thirty-three schools score below 0.1, which indicate these schools are poor with respect to convenience conditions. Of them, there are 17 village schools, 14 township schools, and two urban schools. The spatial distribution map in Figure 3 shows that there is a certain correlation between the level of primary schools' configuration and primary schools' attributes.

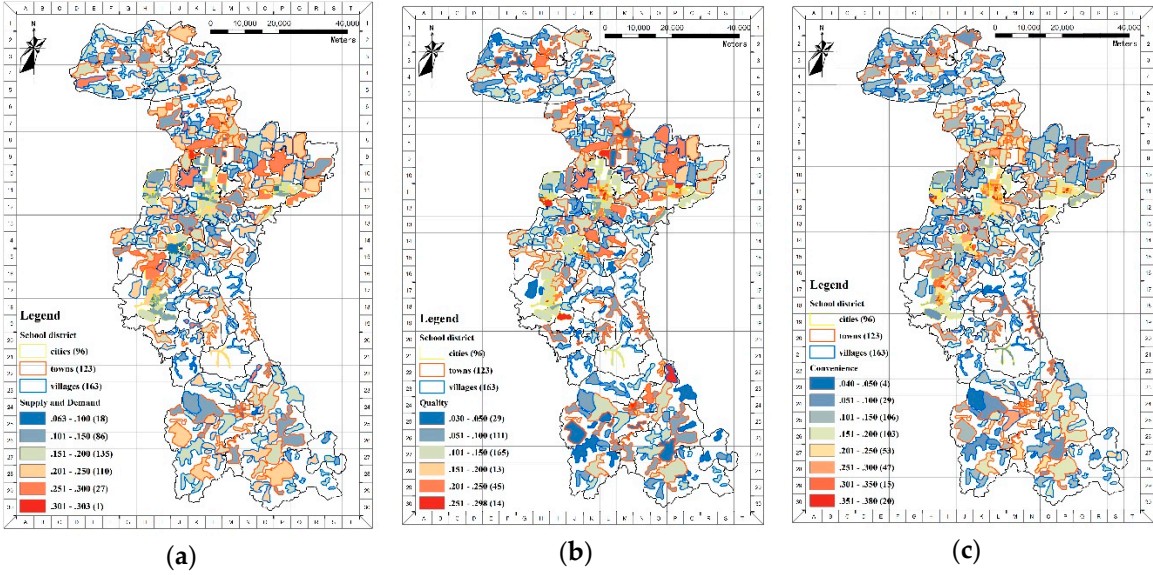

(**a**)　　　　　　　　　　　　(**b**)　　　　　　　　　　　　(**c**)

**Figure 3.** The distribution of three categories scores including (**a**) supply and demand scores; (**b**) quality scores; (**c**) convenience scores in Zibo City.

The spatial distribution profile of the overall score is displayed after the demonstration of the three classification scores. As illustrated in Figure 4, the comprehensive scores calculating from the indicator system of primary school's configuration show that primary schools with low scores, which are displayed in blue and show that the overall configuration conditions are relatively poor. These schools are mainly distributed in villages and some towns. Primary schools with high total scores, which are indicated in red and demonstrate the overall configuration conditions are relatively good. These schools are mainly distributed in urban areas and other towns. The school districts displayed in red and orange are mainly located in the center of five urban districts and three counties. The scores have decreasing trends from the circle center to the edge. The five urban districts are better than three counties. The villages located on the edge of the county have the lowest scores.

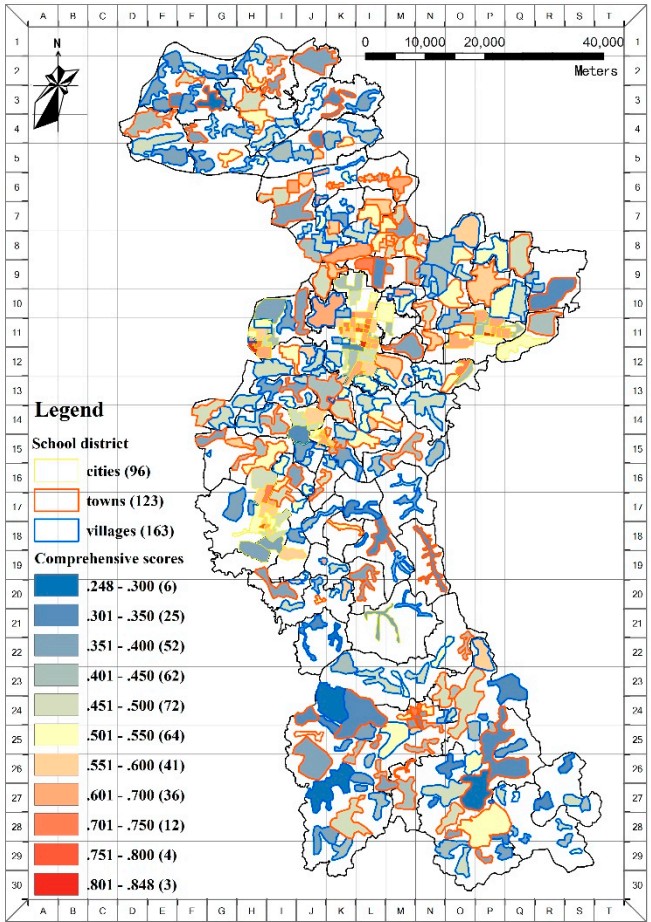

**Figure 4.** The distribution of comprehensive scores in Zibo City.

*5.2. The Results of the Kruskal–Wallis Test*

After checking, the data, which will be analyzed, have passed the four assumptions as follows: Comprehensive score is a dependent variable, which is measured at the continuous level. In addition, the independent variable consists of three categories. Moreover, there is no relationship between the observations in each group. Finally, the distributions in each group have the same shape. Therefore, to further verify the correlation between the level of primary schools' configuration and the attributes of the primary schools, independent sample nonparametric tests were performed on the sample data from the cities, towns, and villages. The characterization values of cities, towns, and villages in the independent sample nonparametric test were 1, 2, and 3, respectively.

Overall scores and category scores are tested, respectively. Table 3 is the summary of the four test results, including overall score test and three-category score tests. It is compiled from the forms generated by SPSS 25 (IBM Corp. Released 2017. IBM SPSS Statistics for Windows, Version 25.0. Armonk, NY: IBM Corp) [50]. The results show that the comprehensive score, quality score, and convenience score tests' significance level is 0.000 < 0.05, and the supply and demand score significance level is 0.001 < 0.05. Therefore, the null hypotheses are all rejected. This also means that the configuration level of primary schools, including overall, supply and demand, quality, and convenience, all show a ternary development. All of them have significant differences in the urban, township, and rural areas.

**Table 3.** The summary of the four Kruskal–Wallis independent sample test results.

| | Null Hypothesis | Test | Sig. | Decision |
|---|---|---|---|---|
| 1 | The distribution of comprehensive score is the same across categories of district. | | 0.000 | |
| 2 | The distribution of supply and demand is the same across categories of district. | Independent-Samples Kruskal–Wallis Test | 0.001 | Reject the null hypothesis. |
| 3 | The distribution of quality is the same across categories of district. | | 0.000 | |
| 4 | The distribution of convenience is the same across categories of district. | | 0.000 | |

Asymptotic significances are displayed. The significance level is 0.05.

Figure 5 is the "Independent-Samples Kruskal–Wallis Test" boxplots, which are produced by SPSS 25 directly. The boxplots reflect the median and distribution of comprehensive, supply and demand, quality, and convenience scores for each group by five statistics, including maximum value, first quartile, median, third quartile, and minimum value. The exceptional data points and outliers can also be displayed in the boxplots, where "o" means outliers, which is 1.5 times bigger or smaller than first quartile and third quartile, and "*" indicates exceptional data points, which is 3 times bigger or smaller than first quartile and third quartile.

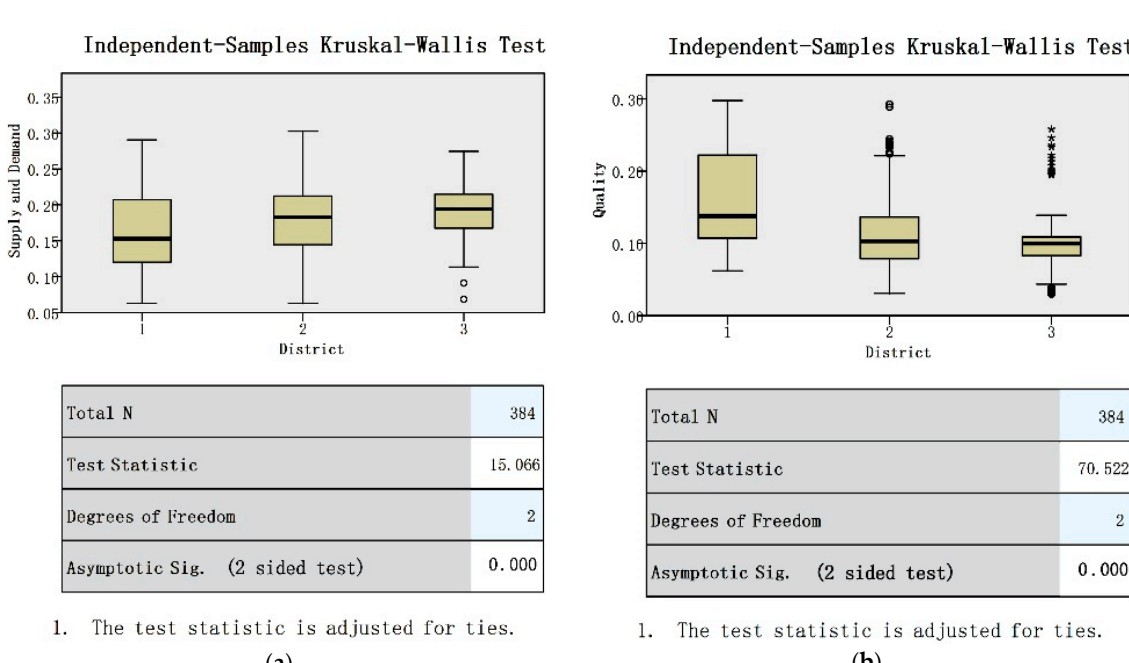

(a)　　　　　　　　　　　　　　　　　　　(b)

**Figure 5.** *Cont.*

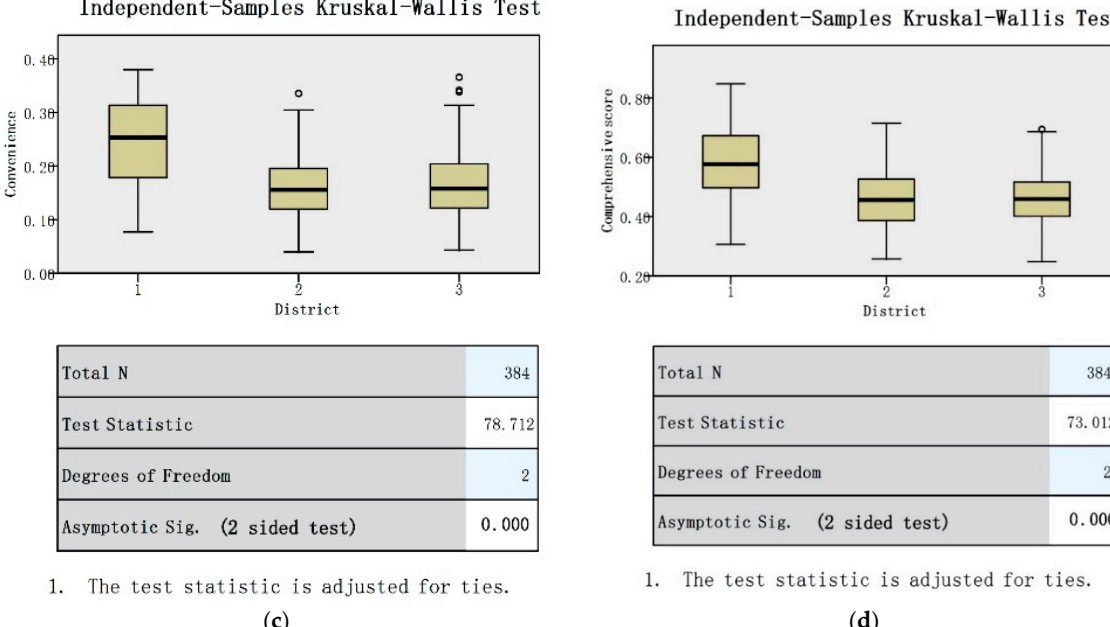

**Figure 5.** Kruskal–Wallis independent sample test boxplots of three categories scores including (**a**) supply and demand score results; (**b**) quality score results; (**c**) convenience score results; (**d**) comprehensive score results.

Figure 5 shows the Kruskal–Wallis independent sample test boxplots of three category scores. It is demonstrated that the supply and demand situation in urban primary schools is worse than in towns and villages in Figure 5a. The four statistics of urban primary schools are the lowest ones except for the maximum value. Four statistics of village schools, including the first quartile, median, third quartile, and minimum value, are the greatest of all three categories. The village category has some outliers, which is 1.5 times smaller than the third quartile because some villages have become hollow villages. The population, especially young and middle-aged laborer's and school-aged children, have been seriously drained. The campus that originally could accommodate multiple classes has only three or fewer classes, resulting in a mismatch between supply and demand. The primary schools in towns have the highest maximum value and the lowest minimum value.

When it comes to the quality scores boxplot in Figure 5b, the urban primary schools still have an absolute advantage. The median of urban schools is much closer to the third quartile than the first quartile. The third quartile of urban schools is higher than the medians of other types. This means the number of urban primary schools with higher scores is more than the lower scores. Both the outliers and the exceptional data points of town and village schools show that the quality scores' distribution of town and village schools is dispersed. For town schools, there are several schools with especially good quality in the center of the towns. For the village schools, the five statistics are low and close to each other. It further confirms that the overall quality of rural primary schools is poor. Additionally, outliers and exceptional data points indicate that there are a certain number of both good and poor quality primary schools.

As for the convenience score boxplot in Figure 5c, similarly, the five statistics for urban primary schools are far ahead of the other types of primary schools. Especially, five statistics illustrate that town primary schools have the worst convenience. The reason is that large-scale rural schools "closure and merger" make town schools serve more and more villages. The outliers of village schools are because some villages preserved the learning centers that serve only one or several villages.

Figure 5d shows the Kruskal–Wallis independent sample test boxplot of comprehensive score results. From Figure 5d, urban primary schools have the obvious highest value of five statistics for overall scores. The third quartile is still higher than town and village's median. Three statistics of

town schools' comprehensive score, including maximum value, first quartile and third quartile are slightly greater than villages', while the other two statistics, median and minimum value, are slightly lower than villages' value. Especially, the village category has some outliers, which is 1.5 times greater than the first quartile; it is relative to those exceptional data points in Figure 5b. Generally, the results analyzed by SPSS correspond with the GIS analysis results. The summary of evaluation results is shown in Table 4.

**Table 4.** The summary of evaluation results.

| Category | Cities | Towns | Villages |
|---|---|---|---|
| Supply and Demand | Poor | Overall average, large interval span | Overall relatively good, only a few are poor |
| Quality | Good | Overall average, some schools good | Poor overall, only a few are good |
| Convenience | Good | Poor | Poor for some schools but good for another schools |

## 6. Discussion and Proposals

As presented in the previous section, the datasets and results provide rich information for identifying the primary school configuration difference in cities, towns, and villages. Based on the analysis results, it is necessary to analyze its drivers and present strategies in sustainable urban development.

As a transitional unit between urban and rural areas, the town is reflected in the configuration of basic education facilities. The town is not just an administrative unit. Towns in China are not only different from the villages, but also different from the cities in terms of economic conditions, population density, and public education funding. Basic education facilities have deviated from the dual development structure and transformed towards ternary development. It is also directly related to county-based policies and encourages the development of small town policies [51].

The root causes of urban primary schools being poor at supply–demand are closely related to the history of the schools' operation, the level of economic development, and the education finance system. Families will inevitably choose quality schools following the principle of maximizing benefits. Therefore, the school choice phenomenon in the inner city will appear [52]. Therefore, real estate developers increase the plot ratio of the old residential area and use high-quality education resources to control housing prices. Due to land property rights, it is difficult to expand primary schools in the old inner city. Therefore, there is bound to be a problem of supply and demand mismatch. In addition to supply and demand issues, it also brings other problems. One of the basic education facilities' public goods attributes, the non-exclusive attribute, is lost and basic education resources are becoming competitive. Then, high-quality education resources face the free-rider problem. For every additional student, the reasonable level of education services will drop slightly. The student–teacher ratio increases. The average attention paid by the teacher decreases. The building area per student declines and the marginal cost increases. In addition, the government pays exclusive costs, transforming from going to a proximity school to the strict division of the school's district. The main fact is that it is difficult to guarantee the fairness of the educational resources' distribution even when the government spends a great deal on monitoring costs.

In the past 20 years China's universities have continued to expand enrolment, which has brought the increasing promotion of the labor force level in the talent market. When the city's labor force is saturated, talent first spills out into the town. For the convenience, in the process of urbanization development, the rural population moved to cities, and with the widespread implementation of the family planning policy, the rural population has decreased significantly; the number of school-aged children has decreased year-by-year. Large-scale rural school closures and mergers make more and more schools run at large scales. Although the Chinese government has improved the hardware

conditions of these schools to a certain extent, the local people in towns and villages are also facing many practical problems, such as the increasing the service radius of schools and the long distances to schools. This problem not only commonly exists in towns but also in some villages. Some schools serving small scales have good convenience, and the convenience of primary schools serving multiple villages has greatly declined.

Most cities of China have just achieved the popularization of compulsory education and the equalization of education funds in the equalization process of basic education facilities. The configuration of basic education facilities is in the transitional stage from guaranteed quantity to guaranteed quality. However, with the promotion of equalization of basic education facilities, although the hardware conditions have certainly improved, the gap in software conditions is gradually widening. This is mainly reflected in the comparison of teachers' quality. Obviously, highly-educated teachers are concentrated in urban areas; what is worse is that village teachers are lacking and becoming older. The imbalanced economic development in cities, towns, and villages has caused a one-way flow of human resources from villages to towns and cities, and the talent drain in underdeveloped rural areas has further exacerbated disadvantage of education facilities. It has formed a vicious circle.

Equal development of basic education facilities is one of the goals of building a sustainable society. Nevertheless, it is not a single indicator's problem rather than a complex problem involving many aspects. Inequality can start before birth, and many of the gaps may compound over a person's life. When that happens, it can lead to persistent inequalities. This can happen in several ways, especially in the nexus among health, education, and parents' socioeconomic status [53]. In the long run, proposals for sustainable society need to be put forward as follows: The development goals of urbanization should be clarified. They are inseparable from the equal development of basic education facilities. For example, how should cities, towns, and villages develop; whether the villages will be replaced by townships decades later; and what is the suitable proportion among cities, towns, and villages? All of these issues will directly decide the development trend and direction of the basic education facilities. Then, the environmental capacity for development and construction land in cities, towns, and villages should be established. Environmental capacity includes plot ratio, population density, and so on, and is closest to the facilities' capacity and service radius. Most importantly, a property rights policy for land redevelopment should be formulated. The integration of the land resources department and the urban planning department further constrained the new development and construction land indicators in China. For built-up land, the adjustment of public-use facility land needs the support of system and policy design. Fourth, the equalization of talent allocation needs to be addressed. The level of teachers is one of the reasons for the uneven development of education facilities. Only the balanced flow of talents can fundamentally solve this problem. Thereby, the essence of the talent's balanced flow is the equalization of economic and social development.

At present, there are at least two proposals for a sustainable society. It is necessary to further promote public education funding and talent support in vulnerable areas through national policies. For example, the government encourages highly-educated people to go to the countryside and assist local education. Furthermore, the policymakers should do their best to ensure that each current school-aged child receive a quality education; for example, keeping necessary learning centers in rural areas. Third, the configuration of basic education facilities needs to consider the development of urban–rural integration; for example, consider the education of off-farm workers' children comprehensively.

## 7. Conclusions

Both the entropy weight and Kruskal–Wallis methods in this study allowed us to delineate the configuration differences of primary schools with different administrative affiliations in China. Different from the well-known urban–rural separation, our findings showed that there was a ternary development of the primary schools' configuration in the cities, towns, and villages.

Through the entropy weight method, it exposed that the comprehensive performance of cities' primary schools was the best one of the three primary school types, but not every aspect was better

than the others. City primary schools were good at teaching quality and home-to-school commuting convenience, but poor at supply–demand conditions. Both increasing students attracted by the fame of the school and limited school size had put the schools' resources in short supply. In addition, the primary schools in the centers of the towns always had better quality, but the large-scale town primary schools were always poorer in convenience because excessive centralized schooling will inevitably lead to a substantial increase in commuting costs. Although the government had greatly improved the material conditions of rural primary schools, this study showed that rural areas remained vulnerable with respect to education development.

Similarly, the results of the Kruskal–Wallis method confirmed the facts above again. However, the results of the Kruskal–Wallis method also presented a great deal of mathematical character of the three data groups. More specifically, both the maximum and minimum showed that the supply–demand conditions in towns presents polarization. The exceptional data points and outliers in the quality performance indicated that the development of rural schools themselves was also extremely uneven. A few outliers in the convenience boxplot also demonstrated that some schools in towns and cities have poor school-to-home convenience.

The ultimate goal of sustainable development is equalization of education facilities. The result of this research shows that there is still a long way to achieve it. Thus, for areas where unsustainable flows exist, we were able to demonstrate and suggest how to maintain or shift the current state in the future. In this way, we were able to understand to what extent the efforts of government departments or other social organizations have paid off, leading us to important information about where to get more material support and where to make institutional changes.

**Author Contributions:** Conceptualization, C.L. and W.S.; methodology, C.L. and W.S.; software, C.L. and W.S.; validation, X.H., Z.L. and W.S.; formal analysis, X.H. and W.S.; investigation, W.S.; resources, Z.L.; data curation, W.S. and Z.L.; writing—original draft preparation, X.H. and W.S.; writing—review and editing, C.L., Z.L., X.H. and W.S.; visualization, Z.L.; supervision, C.L.; project administration, Z.L.; funding acquisition, W.S. All authors have read and agreed to the published version of the manuscript.

**Funding:** This research was funded by National Natural Science Foundation of China (Youth Program), grant number 51808319.

**Conflicts of Interest:** The authors declare no conflict of interest.

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
