# Peer review of "Identifying the Configuration Differences of Primary Schools with Different Administrative Affiliations in China"

_buildings, doi:10.3390/buildings10020033_

Round 1

Reviewer 1 Report

The paper deals with the assessment and comparison the configuration of primary schools in cities, towns and villages area by building an
13 indicator system and using entropy weight method. The paper is well presented and interesting. In my opinion, authors should mark and underline how the work follows the scope of the journal. Moreover, it should be explained if the method can be used as well for different end-use and other case study and how this method can increase the sustainability of the cities. The quality of figures 5 and 6 should be improved. A new section "case study description" should be added, moving the description from "4. Data and implementation" section. Finally, more results in conclusion section should be added.

Author Response

The authors are grateful to the reviewer for their insightful suggestions and constructive comments, which have greatly contributed to improving this paper. The authors have seriously considered all suggestions and have revised the manuscript in accordance with the editor and reviewers’ comments. Please see details of their responses to the editor and reviewers’ comments as follows.

Authors’ responses to Reviewer 1’s comments

Comment:

In my opinion, authors should mark and underline how the work follows the scope of the journal.

Response:

The editors said submissions might be an “advanced approaches in sustainable urban development” in the instruction of the special issue information. My study happened to be an approach providing facts and basis for policymakers to achieve the equity configuration in the sustainable urban development.

I have amended the content to strengthen it in my paper for several times. More specific, one sentence is added in the last sentences of the abstract, in paragraph 2 line 59(revised edition), and in line 110(revised edition), respectively. Moreover, “It is a strategy in sustainable urban development” is  emphasize in line 456(revised edition) of Section 6 discussion and proposal. Finally, in the conclusion it is strengthened again about what needs to be tackled to pursue the Sustainable Development Goals by a final box to correspond with the scope of the journal.

Comment:

Moreover, it should be explained if the method can be used as well for different end-use and other case study and how this method can increase the sustainability of the cities.

Response:

One sentence is added to explain this method can be used as well for other case study in the line 138(revised edition) of Section 3 methodology.

About the question “how this method can increase the sustainability of the cities”, it is describing in line 131(revised edition).

Comment:

The quality of figures 5 and 6 should be improved.

Response:

The figures 5 and 6 are produced by SPSS directly, I noticed that several characters are overlapped together, so I modified them in the software Photoshop.

 Comment:

A new section "case study description" should be added, moving the description from "4. Data and implementation" section.

Response:

I have added the new section “4. case study description”, moving the description from “5.Data and implementation”, and change the title of Section 5 as “Implementation and results”. The previous3.1. Selection of the indicators for the configuration of primary schoolsand “4.1. The scores and spatial distribution of each indicator” are combined together as current “4.2. Selection of the indicators for the configuration of primary schools”.

Comment:

Finally, more results in conclusion section should be added.

Response:

The conclusion is divided into three paragraphs and some information in details is added. The first paragraph is an overall conclusion. The second and third paragraph is focused on the Section 5.1 and 5.2, respectively. In addition, a final box (table) about the indicators is added in the conclusion.

Authors’ own correction

Extra1: the optimization of the paper title

In order to be more concise, the title of the paper is modified as “Identifying the configuration difference of primary schools in different communities of China”

Extra2: 5 important references are added.

Reviewer 2 Report

The paper “Is the configuration of primary schools equity in cities, towns and villages of China? The case of Zibo for the sustainable development proposals” is pretty interesting, nonetheless, some issues arise:

- The abstract does not place the issue to be tackled with this research, nor the outputs.

- Conclusions need to put forward the advances of the research, which means to exactly specify what needs to be tackled to pursue the Sustainable Development Goals (SDGs). A final box (table) about the indicators, the eventual update and their correlation with the SDGs would be really helpful. If not, conclusions seem a bit vague –they are qualitative, whilst the study has been qualitative-, because there is not a qualitative final analysis of what to improve and, with that qualitative summary, to indicate the possible directions to be taken towards future.

- In the methodology section, a brief explanation of why those indicators have been chosen and why others have been discarded is necessary.

- Some references about Chinese laws are needed at the beginning of the paper, in the first paragraphs.

- The English language needs a minor review on phrases such as line 428, …

Author Response

The authors are grateful to the reviewer for their insightful suggestions and constructive comments, which have greatly contributed to improving this paper. The authors have seriously considered all suggestions and have revised the manuscript following the editor and reviewers’ comments. Please see details of their responses to the editor and reviewers’ comments as follows.

Authors’ responses to Reviewer 2’s comments

Comment:

The abstract does not place the issue to be tackled with this research, nor the outputs.

Response:

I have reorganized the abstract.

Comment:

Conclusions need to put forward the advances of the research, which means to exactly specify what needs to be tackled to pursue the Sustainable Development Goals (SDGs). A final box (table) about the indicators, the eventual update and their correlation with the SDGs would be really helpful. If not, conclusions seem a bit vague –they are qualitative, whilst the study has been qualitative-, because there is not a qualitative final analysis of what to improve and, with that qualitative summary, to indicate the possible directions to be taken towards future.

Response:

The conclusion is divided into three paragraphs and some information in details is added. The first paragraph is an overall conclusion. The second and third paragraph is focused on Section 5.1 and 5.2, respectively. Besides, a final box (table) about the indicators is added in the conclusion.

Comment:

In the methodology section, a brief explanation of why those indicators have been chosen and why others have been discarded is necessary.

Response:

Almost a whole paragraph is added to explain why those indicators have been chosen and why others have been discarded is necessary.

Comment:

Some references about Chinese laws are needed at the beginning of the paper, in the first paragraphs.

Response:

Three new references about Chinese laws and policies are added. One specification position is adjusted.

Comment:

The English language needs a minor review on phrases such as line 428, …

Response:

The sentence has been modified and other sentences have been checked and reviewed again.

Authors’ correction

Extra1: the optimization of the paper title

To be more concise, the title of the paper is modified as “Identifying the configuration difference of primary schools in different communities of China”

Extra2: 5 important references are added.

Round 2

Reviewer 1 Report

The paper has been improved. I suggest avoiding many references in the same group (e.g. 9-13) trying to specify the contribution of each cited document. For the figures, I suggest choosing colours with more contrast to make more clear them (e.g.Figure 4). Table 4 should be moved in the results section.

Author Response

The authors are grateful to the reviewer for their insightful suggestions and constructive comments, which have greatly contributed to improving this paper. The authors have seriously considered all suggestions and have revised the manuscript in accordance with the editor and reviewers’ comments. Please see details of their responses to the editor and reviewers’ comments as follows.

Authors’ responses to Reviewer 1’s comments

Comment:

The paper has been improved. I suggest avoiding many references in the same group (e.g. 9-13) trying to specify the contribution of each cited document.

Response:

I have added text description to specify the contribution of each cited document.

Comment:

For the figures, I suggest choosing colours with more contrast to make more clear them (e.g.Figure 4).

Response:

To demonstrate the analysis results more clearly, I have re-edited all of the ArcGIS Figures, re-selected their colours and categorized all analysis data.

Comment:

Table 4 should be moved in the results section.

Response:

Table 4 has been moved in the results section.

Reviewer 2 Report

I want to thank the authors for their modifications, while some parts of the paper have been improved:

The conclusions over the methodology and results sections need to be more quantitatively explicit. This specification intends to denote coherence between the different parts of the paper. Table 4 is not clear to me or at least is not clearly/accurately related to the rest of the analysis exposed in the article. Perhaps a clearer reference of what or how the authors value what is Good or Poor needs to be clearly based on the outputs of the Results section. There are still no indications about which should be the future directions of new research based on the topic of this paper. The English language still needs proofreading.

Author Response

The authors are grateful to the reviewer for their insightful suggestions and constructive comments, which have greatly contributed to improving this paper. The authors have seriously considered all suggestions and have revised the manuscript in accordance with the editor and reviewers’ comments. Please see details of their responses to the editor and reviewers’ comments as follows.

Authors’ responses to Reviewer 2’s comments

Comment:

The conclusions over the methodology and results sections need to be more quantitatively explicit. This specification intends to denote coherence between the different parts of the paper.

Response:

I have re-edited the conclusions.

Comment:

Table 4 is not clear to me or at least is not clearly/accurately related to the rest of the analysis exposed in the article. Perhaps a clearer reference of what or how the authors value what is Good or Poor needs to be clearly based on the outputs of the Results section.

Response:

To indicate what is Good or Poor based on the outputs of the Results section, I have categorized all analysis data, re-edited all of the ArcGIS Figures and explained it in the corresponding paragraph. Table 4 has been moved into the results sections.

Comment:

There are still no indications about which should be the future directions of new research based on the topic of this paper. The English language still needs proofreading.

Response:

The future directions of new research based on the topic of this paper have been added in the last paragraph of the conclusions.
